# What makes music memorable? Relationships between acoustic musical features and music-evoked emotions and memories in older adults

Ilja Salakka[1,2]*, Anni Pitkäniemi[1], Emmi Pentikäinen[1], Kari Mikkonen[3], Pasi Saari[4], Petri Toiviainen[4], Teppo Särkämö[1]

1 Department of Psychology and Logopedics, Music, Ageing and Rehabilitation Team, Cognitive Brain Research Unit, Faculty of Medicine, University of Helsinki, Helsinki, Finland, 2 The Rehabilitation Foundation, Helsinki, Finland, 3 Sentina Ltd, Rajamäki, Finland, 4 Department of Music, Art and Culture Studies, University of Jyväskylä, Jyväskylä, Finland

* ilja.salakka@helsinki.fi

## Abstract

### Background and objectives

Music has a unique capacity to evoke both strong emotions and vivid autobiographical memories. Previous music information retrieval (MIR) studies have shown that the emotional experience of music is influenced by a combination of musical features, including tonal, rhythmic, and loudness features. Here, our aim was to explore the relationship between music-evoked emotions and music-evoked memories and how musical features (derived with MIR) can predict them both.

### Methods

Healthy older adults (N = 113, age ≥ 60 years) participated in a listening task in which they rated a total of 140 song excerpts comprising folk songs and popular songs from 1950s to 1980s on five domains measuring the emotional (valence, arousal, emotional intensity) and memory (familiarity, autobiographical salience) experience of the songs. A set of 24 musical features were extracted from the songs using computational MIR methods. Principal component analyses were applied to reduce multicollinearity, resulting in six core musical components, which were then used to predict the behavioural ratings in multiple regression analyses.

### Results

All correlations between behavioural ratings were positive and ranged from moderate to very high (r = 0.46–0.92). Emotional intensity showed the highest correlation to both autobiographical salience and familiarity. In the MIR data, three musical components measuring salience of the musical pulse (Pulse strength), relative strength of high harmonics (Brightness), and fluctuation in the frequencies between 200–800 Hz (Low-mid) predicted both

**Funding:** Financial support for the work was provided by the Academy of Finland (grants 299044, 305264, 306625, and 327996) and the European Research Council (ERC-StG grant 803466). The funders had no role in study design, data collection and analysis, decision to publish, or preparation of the manuscript. Sentina Ltd provided support in the form of salaries for author K.M., but did not have any additional role in the study design, data collection and analysis, decision to publish, or preparation of the manuscript. The specific roles of these authors are articulated in the 'author contributions' section.

**Competing interests:** The author K.M. is employed by Sentina Ltd. This does not alter our adherence to PLOS ONE policies on sharing data and materials. Sentina Ltd do not profit financially or otherwise from the study results. The authors have declared that no other competing interests exist.

music-evoked emotions and memories. Emotional intensity (and valence to a lesser extent) mediated the predictive effect of the musical components on music-evoked memories.

## Conclusions

The results suggest that music-evoked emotions are strongly related to music-evoked memories in healthy older adults and that both music-evoked emotions and memories are predicted by the same core musical features.

## Introduction

Ubiquitous to human culture throughout history [1], music is a unique and complex phenomenon, both regarding its rich acoustic structure, comprising multiple sound features organized around hierarchical principles referred to as musical syntax, and the parallel perceptual, cognitive, and emotional processes that arise when we experience music [2, 3]. Structurally, music comprises several features of varying levels of abstraction, including low-level music features, such as loudness (intensity of sounds), pitch (organization of sounds along a scale from low to high), and timbre (quality that differentiates sounds with same pitch and loudness), and high-level music features, such as tonality (relative structure of pitches that form musical keys, chords and melodies) and rhythm (organization of sound events in time). The perception of these features is acquired largely automatically, through implicit learning by exposure to certain kind of music during development and through enculturation [4–8]. With the development of advanced computational analysis of audio signals, a large number of these features can be automatically extracted from music using methods collectively known as Music Information Retrieval (MIR) [9]. Coupling the time course of musical features extracted with MIR with the time series of blood oxygenation level-dependent (BOLD) signal measured with functional magnetic resonance imaging (fMRI) while listening to the musical piece have revealed that the processing of musical timbre, rhythm, and tonality are associated with the large-scale activation in temporal (superior and middle temporal gyri, insula), frontal (superior and middle frontal gyri, cingulate gyrus, precentral gyrus), parietal (inferior parietal gyrus, precuneus, postcentral gyrus), and cerebellar regions [10, 11].

While music can be viewed as structured sound comprising of different acoustic components, the primary reason why we humans listen to music is its ability to evoke strong and vivid emotions and influence mood [12]. In everyday life, music is most commonly utilized in emotional self-regulation [13], and music also has extensive therapeutic value in alleviating stress, anxiety, and depression [14, 15]. While some aspects of music cognition, such as aesthetic responses to music [16], seem to be culture dependent, there seems to be some universal component in music-evoked emotions [17, 18]. Music-evoked emotions can be construed along three core dimensions: valence (continuum from an unpleasant to pleasant emotional experience), arousal (continuum from low to high level of arousal), and intensity (continuum from weak to strong emotional experience) [19]. By combining behavioural emotion ratings of music and musical features extracted with MIR, it is possible to map which musical features best represent musical emotions, both perceived and experienced. Previous MIR studies suggest that the valence and arousal induced by music are associated with a combination of loudness (RMS energy) and tonal (key mode, spectral entropy), rhythmic (pulse clarity, tempo), and timbral (spectral flux) features [20–22]. It should be noted that this linkage applies to subjective ratings of valence / arousal after exposure to a song while any sudden changes in

musical structure during exposure may elicit momentary changes in valence / arousal and associated physiological responses (e.g. chills) [23]. Recently, Singer et al. [24] explored the linkage between music-evoked emotions, musical features, and brain activation utilizing a Dynamic Common Activation (DCA) analysis that combined fMRI BOLD data during music listening with both MIR data of the musical features and continuous emotion rating data of the experienced valence and arousal of the song. This revealed a strong association between music-induced emotionality and DCA modulation specifically in a limbic network comprising for example the amygdala, the hippocampus, and the orbitofrontal cortex [24], which have previously been identified as core regions underlying the emotional experience of music [25, 26]. Interestingly, the link between limbic network activation and music-induced emotionality was found to be mediated especially by temporal musical features (beat strength and tempo), suggesting that the temporal regularities of music play a key role in emotional response to music in limbic brain regions.

In addition to valence, arousal, and intensity of emotions, the emotional experience evoked by a particular song is also strongly influenced by listener's familiarity with it [27]. The familiarity of music and the affective responses to music seem to be closely interlinked: when hearing a song, its familiarity leads to expectations about the structure of the song—and to the preparation of emotional neuronal networks by expectation—leading to anticipatory arousal and possibly also to stronger experienced emotions [27]. Likewise, when appraising music, familiarity and "liking" typically go together: as a song becomes more familiar through repeated exposure, it is rated higher in emotional intensity [27] and valence [28], even in atonal music [29]. At the neural level, listening to familiar compared to unfamiliar music leads to stronger activation in emotion-related and reward-related limbic and paralimbic regions, including the anterior cingulate, the amygdala, and the striatum [30], together with other memory-related and motor-related regions, such as the inferior and superior frontal gyri and the cerebellum [31, 32].

In addition to familiarity, music can evoke autobiographical memories, which are event-specific to lifetime periods, or their content, comprising either semantic or episodic knowledge [33]. Music-evoked autobiographical memories (MEAMs) are autobiographical memories specifically elicited when hearing music from one's past and they are typically coupled with the evocation of emotions, most often positive, which are experienced strongly [33]. MEAMs, like all involuntary autobiographical memories, are also likely to be stronger and more specific than consciously recalled memories [34]. The reported familiarity of music and MEAMs are linked to each other to an extent, with small [35] or moderate [33, 36] correlations, but this relationship is not clear-cut, as not all familiar songs elicit MEAMs and sometimes even unfamiliar songs can elicit MEAMs, possibly through associations with the musical genre of the song [33]. The key brain areas linked to the processing of the autobiographical salience of music are the dorsomedial prefrontal cortex (dmPFC) [35, 37] and the anterior cingulate [38]. These structures have found to be relatively spared in Alzheimer's disease [38], which may explain why even people with late-stage dementia are still able to recall familiar songs and memories associated with them [34]. Also in normal aging, there seems to be age-related shift in the memory and emotion mechanisms underlying the familiarity of music, as it is associated more with the enhancement of memory detail in young adults and affective positivity in older adults [35]. The autobiographical saliency of the music has also been reported to be highest in songs popular during the teenage years of the listener [39].

In summary, emotions are probably one key component in the ability of music to elicit so strong memories [36, 40]. In general, a strong emotional experience creates strong memories [41] and predicts almost all qualities of autobiographical memories better than the valence or the age of the memory [42]. Accordingly, songs that trigger more intense emotions also tend

to elicit more intense MEAMs [36], but also valence is likely to play a role in the formation of MEAMs [33, 43]. As reviewed above, previous studies have provided evidence that music-evoked emotions are connected to the experience of familiarity of music and MEAMs and that music-evoked emotions are linked to specific musical features uncovered with MIR. However, we do not know whether familiarity and MEAMs are directly linked to musical features and if so, is this linkage different than the linkage between music-evoked emotions and musical features. Using a combination of behavioural ratings of 140 song excerpts ranging from traditional folk music to popular music from the 1950s to 1980s by a large sample (N = 113) of healthy older adults and the musical features of the songs extracted with MIR software, the present study sought to (i) explore the relationship between subjective music-evoked emotions (valence, arousal, emotional intensity) and memories (familiarity, autobiographical salience), (ii) determine how musical features predict the subjective experience of music and if this differs for music-evoked emotions and memories, and (iii) establish if the relationship between musical features and music-evoked memories is mediated by music-evoked emotions.

## Methods

### Subjects

The subjects were 113 healthy older adults [86 females; age: mean = 70.7 years, SD = 5.4 years, range 60–86; education level (ISCED, 8-point scale): mean = 4.8, SD = 2.0, range 1–8] from the Helsinki metropolitan area who were participating in an ongoing study on the neurocognitive effects of senior choir singing (78 subjects were amateur choir singers and 35 were non-singers). The subjects were recruited from the Adult Education Centers of the Cities of Helsinki, Espoo, and Vantaa and from different senior citizens' associations and independent choirs through presentations, flyers, and e-mail advertisements. All subjects were Finnish-speaking, and had no history of neurological (e.g., dementia, stroke) or psychiatric (e.g., schizophrenia, bipolar disorder) disorders. The study was approved by the Ethical Review Board in the Humanities and Social and Behavioural Science of the University of Helsinki, and all participants gave written informed consent.

### Procedure

The subjects performed an old-time music rating task (referred to hereafter as OTMR) in which they were asked to listen to and rate altogether 70 song excerpts (see Stimuli). In order to collect data from sufficiently large number of songs without making the task too long and tiresome, the subjects were divided to two groups (A and B) matched for age, gender, and choir singing background, and the song excerpts (140 in total) were equally divided to these two groups (70 in each), matched for song genre and era. The OTMR was implemented as a web browser application created specifically for this study in collaboration with Sentina Ltd, a Finnish company specializing in the design of audio material for recreation and rehabilitation in old-age care. A new module which allowed the structural implementation of the OTMR and data reporting was created on Sentina's cloud service. The subjects were able to do the whole task in any place or time they wanted, using either their own computer or a tablet computer set up and lent by the researchers. The use of headphones or external speakers was recommended.

The OTMR started with a short practice session, in which the subjects were able to test the user interface and set the volume to a comfortable but clearly audible level. After this, the main task followed during which each subject listened to 70 song excerpts. Each excerpt played automatically through once before subjects were able to give rating answers (using a 5-point Likert scale) to five questions on how they experienced the song in terms of (1) Valence (How

pleasant did you find the song? Rating: very unpleasant—very pleasant), (2) Emotional intensity (How strong emotions did the song evoke? Rating: no emotions at all—very strong emotions), (3) Arousal (How did the song affect your arousal state? Rating: decreased arousal significantly—raised arousal significantly), (4) Familiarity (How familiar was the song to you? Rating: not familiar at all—very familiar), and (5) Autobiographical salience (How much personal memories did the song evoke? Rating: no personal memories at all—significant amount of personal memories). The questions were presented one at a time and each had to be answered before continuing to the next. The subjects had the opportunity to listen to the excerpt as many times as they wanted during answering. While answering the questions, the subjects also had a voluntary option to share any memories evoked by the songs, either through a text writing or audio recording interface built into the application (this qualitative data is not included in the present study). The average time for completing the task was 2.5 h.

## Stimuli

The preparation of the song stimuli started by manually searching the archives of main Finnish radio channels to identify a total of common 225 songs comprising traditional (folk) songs as well as songs from 1950s to 1980s of different musical genres (popular, rock and jazz music) and languages (Finnish or English). This song pool (and the OTMR application) was then piloted in 11 healthy older adults, and based on the pilot data, songs with extremely high or low familiarity and autobiographical salience ratings were excluded and two lists (A and B) of 70 song excerpts with relatively balanced song familiarity and autobiographical salience were created for the final OTMR task. The 70 songs in each list comprised 10 folk songs and 15 songs from each four decades (1950s, 1960s, 1970s, 1980s). The full song list is presented in S1 Table. All the audio files were in MP3 format. For each song, an excerpt of the song was selected to represent the most characteristic and well-known part of the song (e.g., the chorus part). The excerpts were on average 30 sec long (range 18–37 sec), which is a typical length in these kinds of experiments [20]. Half sine wave fade-ins (1 sec) and fade-outs (3 sec) were added to each excerpt to make the listening experience smooth.

## Musical feature extraction

The musical features used in the statistical analyses (see below) were automatically extracted using MIRToolbox 1.7 software [44, 45] running in MATLAB version R2018a. The default sampling rate (44100 Hz), which is typical for MP3 files and covers all audible frequencies, was used for the *miraudio* command of MIRToolbox [45]. Frame lengths of 0.025 sec and 3 sec were used in extracting the short-term and long-term features (see below), respectively, which is common in MIR studies [10, 46]. For every extracted feature, an overlap of 50% was used for frame decomposition. In data analysis, the means of each musical feature were computed from the values across frames and then pooled across songs to form the full data.

As briefly summarized in Table 1, a set of 18 short-term and 6 long-term musical features was used in the MIR analysis. More thorough technical descriptions of the features can be found from [45]. These features were chosen to represent different core aspects of music, following what has been used in previous MIR studies [10, 20–22], including timbre, tonality, temporal, and other musical features [45, 47–52].

**Timbre features** tell about the"quality" or"texture" of a sound. *Attack time* is the time from the start of a sound to the peak of its amplitude, which typically varies between instruments (e.g., shorter in percussion instruments, longer in wind instruments). *Spectral centroid* represents a central tendency measure of the spectrum in music, which is higher when there is more energy in the high notes or overtones. *Spectral spread* describes the standard deviation around

**Table 1. Chosen MIR features divided by corresponding time window.**

| Short-term features | Description |
|---|---|
| Attack time | Time from the start of the sound to the first peak of the sound's amplitude. |
| Spectral centroid | The centre of gravity of spectral energy i.e. the weighted mean of the frequencies present in the signal. |
| Spectral spread | The density of power spectrum around the centroid. |
| Spectral flux | The average change in the shape of the spectrum between subsequent frames, representing fast changes in timbre. |
| Sub-band fluxes 1–10 | The spectral flux in limited bandwidths, filtered in 10 different octave-size bands: Sub-band 1 (0–50 Hz), Sub-band 2 (50–100 Hz), Sub-band 3 (100–200 Hz), Sub-band 4 (200–400 Hz), Sub-band 5 (400–800 Hz), Sub-band 6 (800–1600 Hz), Sub-band 7 (1600–3200 Hz), Sub-band 8 (3200–6400 Hz), Sub-band 9 (6400–12800 Hz), Sub-band 10 (12800–25600 Hz) |
| Roughness | A measure of a dissonance obtained through summation of dissonance between all pairs of spectral peaks. |
| Flatness | A measure for smoothness of the spectrum, calculated as a ratio between the geometric mean and the arithmetic mean. |
| Spectral entropy | The so-called Shannon entropy of the spectrum. |
| RMS Energy | Root-mean-square of the amplitude |
| **Long-term features** | **Description** |
| Key clarity | The key strength of the best fitting key, computed from the chromagram*. |
| Mode | A difference between the best fitting major key and the best fitting minor key, computed from the chromagram*. |
| Pulse clarity | A measure of rhythmical clarity computed from an autocorrelation function that estimates the strength of the beats in music. |
| Fluctuation centroid | The geometric mean of the fluctuation spectrum. |
| Fluctuation entropy | Entropy of the fluctuation spectrum. |
| Novelty | A measure of musical expectancy computed by convolving the self-similarity matrix obtained from the spectrogram along its diagonal with a Gaussian checkerboard kernel [53] and computing the mean across time series. |

*Chromagram refers to the distribution of energy on different pitch frequencies corresponding to chromatic scale [9]. Key strength was computed by correlating the chromagram with tone stability profiles representing the 24 keys (12 major, 12 minor; [54]) and taking the maximal correlation.

the spectral centroid. *Spectral flux* is a measure of fast changes in the timbre. *Sub-band fluxes* correspond to the spectral flux in limited bandwidths, filtered in 10 octave-size bands. *Roughness* is a psychoacoustic feature measuring the sensory dissonance of the sound, with higher values indicating more frequencies with short perceptual distance from each other. The final two features were measures of noise: *spectral flatness* describes the smoothness of the spectral distribution whereas *spectral entropy* refers to the so-called Shannon entropy describing how much information the song spectrum contains.

**Tonality features** refer to the aspects representing the musical scales and dominant notes used in the song, including harmonic structure, following the tonal structure of Western music. *Key clarity* represents the strength of the best fitting key for a song, with higher values indicating that the key in which the song is played can be identified clearly. *Mode* represents the degree of majorness or minorness and is obtained by subtracting the strength of the best fitting minor key from that of the best fitting major key.

**Temporal features** refer to time-related aspects of the songs. *Pulse clarity* is a measure of the salience of basic beat in music. *Fluctuation centroid* represents the average frequency of

rhythmical periodicities and *fluctuation entropy* represents rhythmic complexity; together, these two features measure how rhythmic periodicity and diversity affect listeners. Tempo was not included as a feature due to the high tempo variance in some songs.

**Other musical features** are the features that do not fit the above categories. *Novelty* measures how much similarity or dissimilarity there is in the music at different temporal locations and therefore indicates the level of musical expectancy or novelty. *RMS energy* relates to the total magnitude of the audio waveform, and roughly represents the loudness of a song.

## Statistical analyses

Statistical analyses and data handling were carried out with R-language version 3.3.3 [55] in RStudio environment version 1.1.463. Distributions were examined visually and with Kolmogorov-Smirnov test using the R-package *nortest* [56]. The distributions of the behavioural rating scores and the MIR features were all approximately normal, with the exception of Familiarity, which showed high skewness (-1.64) and kurtosis (5.09). The Familiarity scores were transformed by reversing and inverting, as suggested by Tabachnick and Fidell [57], which improved the distribution (skewness = -0.47, kurtosis = 2.28). Multicollinearities were assessed with the R-package *mctest* [58] using VIF and tolerance factors. Principal component analyses were carried out for reducing multicollinearity and parallel analysis was used to test for the optimal amount of components. Both were carried out using the R-package *psych* [59]. In regression analyses, the regression models were created using forward stepwise regression, based on smallest p-value. Testing for the regression coefficients in multiple regression was done with the core R packages [55]. A conservative p-value threshold of 0.005 was adopted in order to increase the replicability of statistically significant findings [60].

## Results

### Relationship between behavioural ratings of music-evoked emotions and memories

The correlations (Pearson, two-tailed) between the five rating scores of music-evoked emotions and memories are shown in Fig 1. All correlations were positive and highly significant (p < 0.001). The correlations were high between the three emotion ratings (Emotional intensity—Valence: r = .91, Arousal—Valence: r = .80; Emotional intensity—Arousal: r = .73) and between the two memory ratings (Autobiographical salience—Familiarity: r = .84). Regarding the linkage between memory and emotion ratings, both Autobiographical salience and Familiarity had the highest correlation to Emotional intensity (r = .92 and r = .71, respectively), followed by Valence (r = .77 and r = .56) and then Arousal (r = .65 and r = .46). Notably, the correlations to the three emotional ratings were 30–41% higher for Autobiographical salience than for Familiarity.

### Relationship between MIR features

The correlations (Pearson, two-tailed) between the 24 MIR features of the song stimuli are shown in S2 Table. There were high correlations between many of the musical features. As high correlations between predictors can lead to multicollinearity in regression analyses, appropriate multicollinearity measures were performed. The variance inflation factor (VIF) values of the musical features ranged from 1.23 to 109.15 and the tolerance values from 0.009 to 0.814. Given that some of the musical features had too high VIF and low tolerance values for regression analyses, PCA was carried out to reduce multicollinearity, as suggested by Eerola et al. [61].

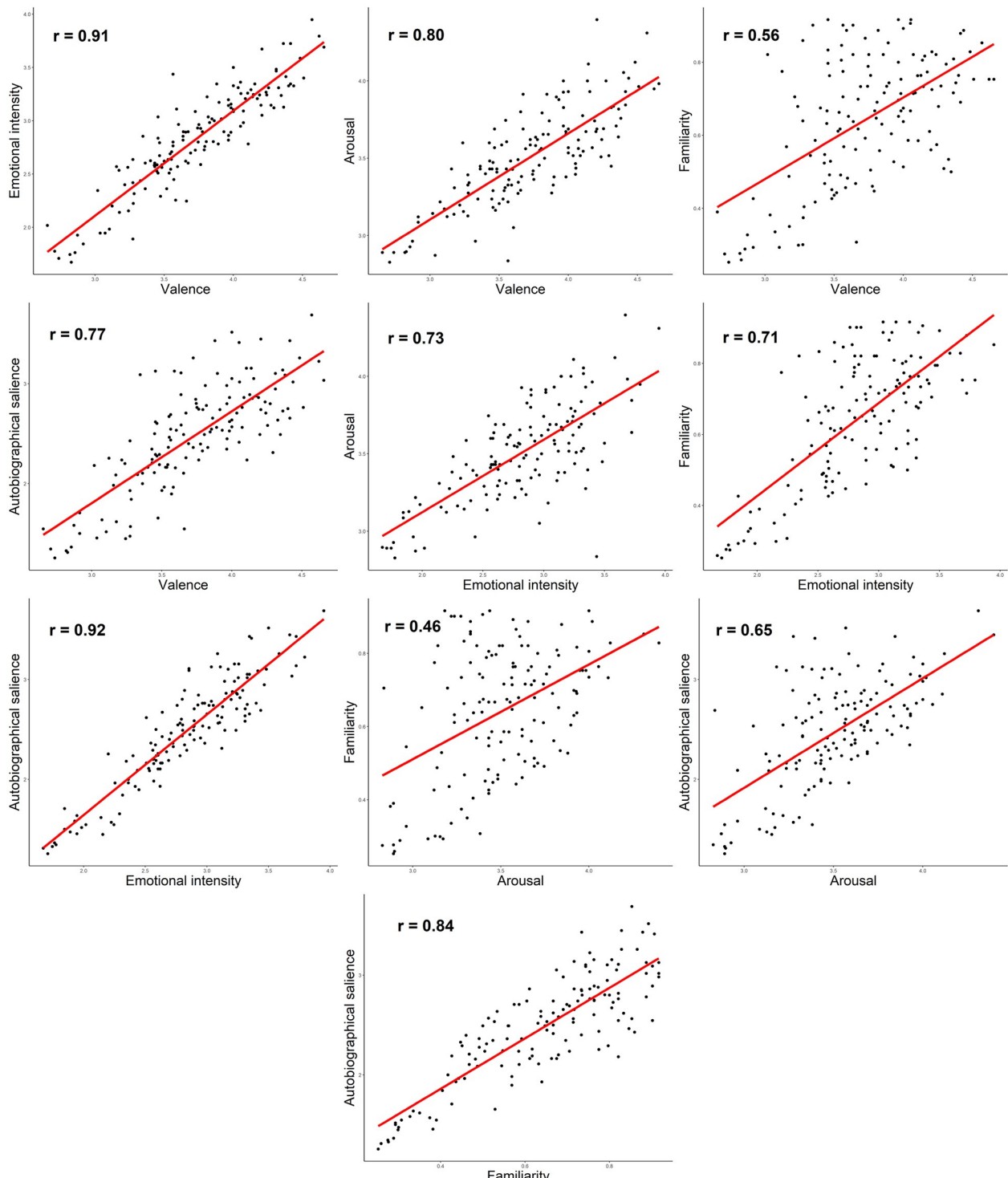

**Fig 1. Relationships between the behavioural ratings of the songs.** p < 0.001 in all pairwise correlations.

**Table 2. Six component PCA solution for musical features using varimax rotation.**

| Musical feature | Musical component | | | | | | Communality |
|---|---|---|---|---|---|---|---|
| | *Brightness* | *High-mid* | *Pulse strength* | *Low-mid* | *Rhythmic clarity* | *Novelty* | |
| Spectral Spread | **.92** | -.09 | .25 | -.04 | .05 | -.01 | .92 |
| Spectral Centroid | **.91** | .25 | .23 | -.06 | .13 | .05 | .96 |
| Flatness | **.89** | -.02 | .34 | -.13 | -.02 | -.08 | .93 |
| Sub-band Flux 10 | **.89** | .20 | .20 | .13 | .07 | -.01 | .88 |
| Sub-band Flux 9 | **.84** | .35 | .24 | .09 | .08 | .05 | .90 |
| Spectral Entropy | **.77** | .47 | .26 | -.08 | .00 | -.02 | .88 |
| Sub-band Flux 7 | .25 | **.83** | .17 | .25 | .07 | .07 | .84 |
| Sub-band Flux 6 | .26 | **.78** | -.01 | .25 | -.12 | -.05 | .76 |
| Roughness | .36 | **.74** | .08 | .33 | .07 | .01 | .79 |
| Sub-band Flux 8 | .43 | **.68** | .25 | .13 | .17 | .09 | .77 |
| Mode | -.24 | **.64** | .17 | -.12 | -.06 | -.02 | .52 |
| Sub-band Flux 2 | .43 | .15 | **.84** | .02 | .05 | .04 | .92 |
| Sub-band Flux 1 | .36 | .16 | **.80** | .03 | -.03 | .14 | .81 |
| Sub-band Flux 3 | .33 | .12 | **.72** | .37 | -.02 | -.13 | .80 |
| Pulse Clarity | .40 | .28 | **.64** | -.15 | .24 | -.24 | .77 |
| Sub-band Flux 4 | -.04 | .04 | .03 | **.88** | -.01 | -.15 | .80 |
| RMS Energy | .25 | .37 | .33 | **.71** | -.05 | .02 | .81 |
| Sub-band Flux 5 | -.11 | .44 | -.32 | **.65** | -.05 | -.09 | .74 |
| Attack Time | .24 | -.06 | -.18 | **-.62** | **.50** | -.08 | .74 |
| Fluctuation Centroid | .04 | .09 | .06 | -.04 | **.92** | .07 | .87 |
| Fluctuation Entropy | -.26 | -.01 | -.19 | .31 | **-.66** | .28 | .71 |
| Key Clarity | .18 | .43 | .14 | -.19 | **-.52** | -.27 | .62 |
| Novelty | .03 | .03 | .00 | -.15 | .00 | **.88** | .80 |
| Proportion variance | .25 | .17 | .14 | .12 | .09 | .05 | |
| Cumulative variance | .25 | .42 | .56 | .67 | .76 | .81 | |
| Proportion explained | .31 | .21 | .17 | .15 | .11 | .06 | |

Loadings greater than or equal 0.5 are shown in bolded.

A PCA model with six components was found to produce the best fit for the data, explaining 81% of the variance, and yield most meaningful and interpretable components (see Table 2). Of the musical features, spectral flux was found to load similarly as the sub-band fluxes and was therefore dropped from the model. The six PCA components (hereafter referred to as *musical components*) were labelled as Brightness, High-mid, Low-mid, Pulse strength, Rhythmic clarity, and Novelty. **Brightness** refers to greater relative amount of higher frequencies in music and had highest loadings from variables that model high frequencies (Spectral centroid, Sub-band fluxes 9–10 with frequencies 6400–25600 Hz), noise (Flatness, Spectral entropy), and spread of frequencies across the spectrum (Spectral spread). **High-mid** had highest loadings from the higher mid-level spectral fluctuation sub-bands (Sub-band fluxes 6–8 with frequencies 800–6400 Hz). **Low-mid** had highest loadings from the lower mid-level spectral fluctuation sub-bands (Sub-band fluxes 4–5 with frequencies 200–800 Hz) as well as from Loudness (RMS Energy) and Attack time, which are likely linked to the sound of percussion instruments typical of this frequency range [62]. **Pulse strength** had the highest loadings from the lowest spectral fluctuation sub-bands (Sub-band Fluxes 1–3 with frequencies 0–200 Hz) and Pulse clarity, which together provide the rhythmic feeling in music [63]. **Rhythmic clarity** had the highest loadings from Fluctuation centroid and Fluctuation entropy,

**Table 3. Regression models for ratings predicted by musical components.**

| Rating variable | Musical component | β | p | R² | F (p) |
|---|---|---|---|---|---|
| Valence | Pulse strength | -0.150 | < .0001 | .254 | 15.41 (p < .0001) |
| | Low-mid | -0.119 | .0005 | | |
| | Brightness | -0.121 | .0004 | | |
| Emotional intensity | Pulse strength | -0.204 | < .0001 | .329 | 22.23 (p < .0001) |
| | Brightness | -0.138 | < .0001 | | |
| | Low-mid | -0.130 | .0002 | | |
| Arousal | Pulse strength | -0.077 | .0032 | .061 | 8.99 (p = .003) |
| Familiarity | Pulse strength | -0.099 | < .0001 | .455 | 37.81 (p < .0001) |
| | Brightness | -0.056 | < .0001 | | |
| | Low-mid | -0.038 | .0010 | | |
| Autobiographical salience | Pulse strength | -0.265 | < .0001 | .411 | 31.67 (p < .0001) |
| | Brightness | -0.183 | < .0001 | | |
| | Low-mid | -0.115 | .0013 | | |

β = unstandardized regression coefficient, p = p-value of single variable, R² = coefficient of determination, F (p) = F-value of the model (p-value in the brackets).
Degrees of freedom for all models are 3 and 136 except for Arousal which are 1 and 138.

which are both rhythmic features. **Novelty** had the highest loading from Novelty. To clarify these abstract concepts, examples of songs with the highest PCA scores in for each component are listed in S3 Table.

## Predicting music-evoked emotions and memories with musical components

The results of the regression analysis are summarized in Table 3. All the five rating scores were successfully predicted by a select combination of the musical components, explaining 6.1%–45.5% of the variance. Pulse strength, Low-mid, and Brightness emerged as significant predictors of Valence, Emotional intensity, Familiarity, and Autobiographical salience, however with different weightings for each rating score. Arousal was predicted only by Pulse strength. Notably, all the regression coefficients (β) in the models were negative, indicating that higher Valence, Arousal, Emotional intensity, Familiarity, and Autobiographical salience were explained by lower pulse strength, narrower frequency spectrum (especially from the high end), and less frequency fluctuation in the lower middle range. Bivariate scatterplots between single musical components and rating variables are shown in Fig 2.

## Mediative effects of emotions on predicting music-evoked memories with musical components

Given the significant correlation of Autobiographical salience to Emotional intensity (r = .92), Valence (r = .78), and Arousal (r = .65), we sought to determine whether these three emotional variables would mediate the predictive effect of musical components on Autobiographical salience. For this purpose, we re-performed the regression analysis of Autobiographical salience using the following three-step procedure: (1) entering the three significant MIR components (Pulse strength, Brightness and Low-mid) to the model, (2) adding the three emotional variables (Emotional intensity, Valence, and Arousal) to the model one at a time, and (3) noting changes in the regression coefficients and p-values of the MIR components after each emotion variable was added. The mediative effects were defined as partial (small change in regression coefficient and retained significance of p-value) or full (large change in

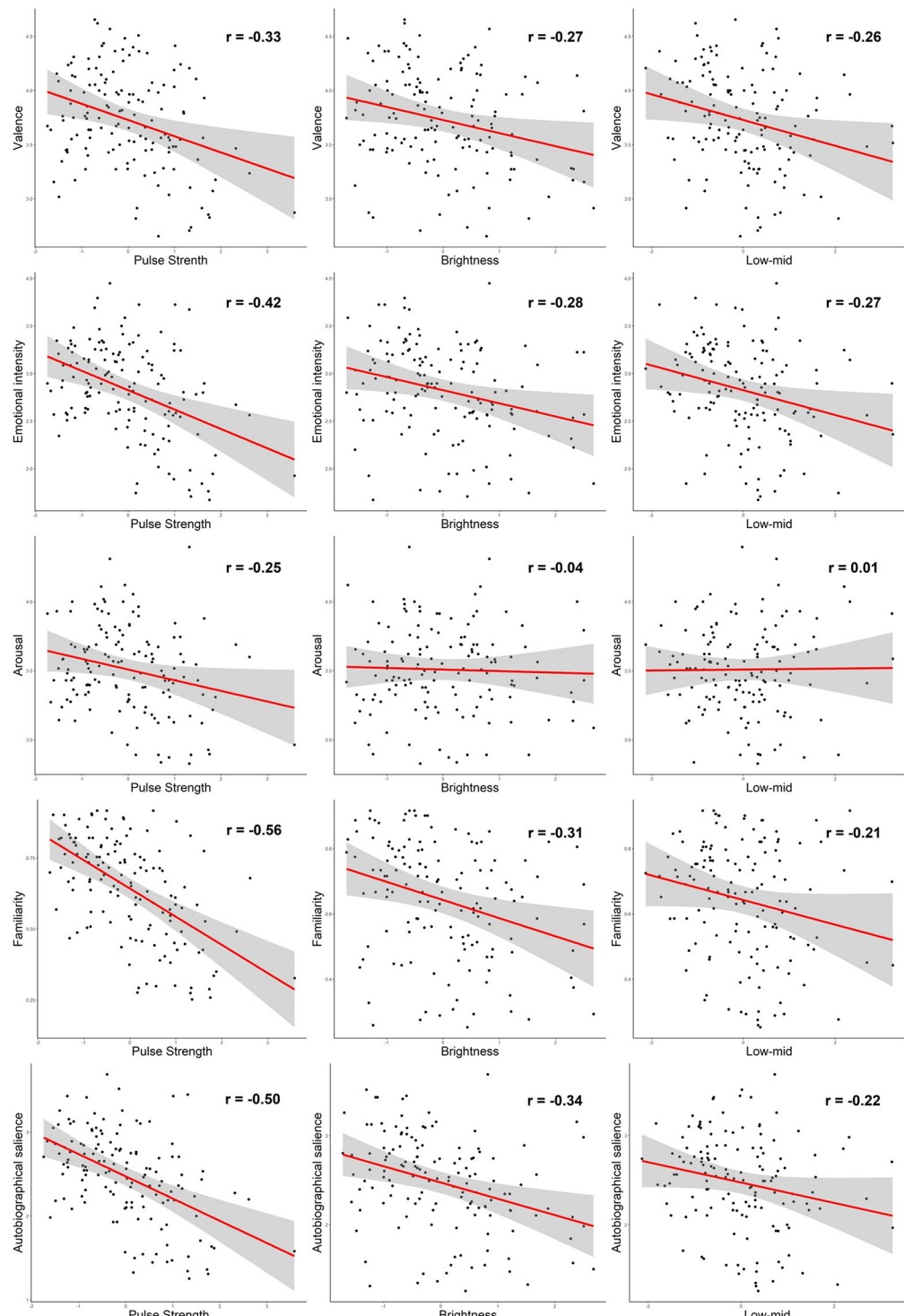

**Fig 2. Regression models for ratings predicted by single musical components.** Scatterplots and regression lines for single explaining variables with standard error confidence interval of level 0.995.

**Table 4. Bivariate correlations, regression coefficients, and p-values of autobiographical salience mediation models for each emotion variable.**

| Variables | Autobiographical salience | Emotional intensity | β | p | $R^2$ | F (p) |
|---|---|---|---|---|---|---|
| Emotional intensity | .92 | | 0.905 | < .0001 | .867 | 219.60 (< .0001) |
| Pulse strength | -.50 | -.42 | -0.081 | < .0001 | | |
| Brightness | -.34 | -.28 | -0.058 | .001 | | |
| Low-mid | -.22 | -.27 | 0.003 | .85 | | |
| | Autobiographical salience | Valence | β | p | $R^2$ | F (p) |
| Valence | .78 | | 0.732 | < .0001 | .694 | 76.69 (< .0001) |
| Pulse strength | -.50 | -.33 | -0.156 | < .0001 | | |
| Brightness | -.34 | -.27 | -0.095 | .0005 | | |
| Low-mid | -.22 | -.27 | -0.028 | .30 | | |
| | Autobiographical salience | Arousal | β | p | $R^2$ | F (p) |
| Arousal | .65 | | 0.930 | < .0001 | .691 | 75.62 (< .0001) |
| Pulse strength | -.50 | -.25 | -0.193 | < .0001 | | |
| Brightness | -.34 | -.04 | -0.172 | < .0001 | | |
| Low-mid | -.22 | -.01 | -0.118 | < .0001 | | |

β = unstandardized regression coefficient of a single predictor in multiple regression model, after controlling for other variables, p = p-value of a single variable, $R^2$ = coefficient of determination of the model, F (p) = F-value of the model (p-value of the model in the brackets). Degrees of freedom for all models are 4 and 135.

regression coefficient and abolished significance of p-value). As shown in Table 4, Emotional intensity had a partial mediative effect on Pulse strength (from β = -0.27, p < .0001 to β = -0.08, p < .0001) and Brightness (from β = -0.18, p < .0001 to β = -0.06, p = .0014), and a full mediative effect on Low-mid (from β = -0.12, p = .0013 to β = 0.003, p = 0.850). Similarly, Valence had a partial mediative effect on Pulse strength (from β = -0.27, p < .0001 to β = -0.16, p < .0001) and Brightness (from β = -0.18, p < .0001 to β = -0.10, p = .0005) and a full mediative effect on Low-mid (from β = -0.12, p = .0013 to β = -0.03, p = .297). Arousal showed a partial mediative effect only on Pulse strength (from β = -0.27, p < .0001 to β = -0.19, p < .0001). Notably, there was a very similar pattern of mediative effects of the emotion variables on the predictive effect of musical components on Familiarity (see S4 Table).

## Discussion

This study sought to explore the relationship between subjective music-evoked emotions and memories and determine how they are predicted by the musical features of the songs in a large sample of healthy older adults, utilizing a combination of behavioural song ratings and MIR analysis. Our main findings were that (i) music-evoked emotions (especially emotional intensity and valence) were strongly related to music-evoked memories (autobiographical salience and familiarity), (ii) both music-evoked emotions and memories were predicted by a core set of three musical components derived from the MIR features (pulse strength, brightness, and low-mid), and (iii) music-evoked emotions (especially emotional intensity) mediated the prediction of music-evoked memories by the musical components.

The behavioural results from the correlation analyses indicated that there was a very strong association between the emotions and the memories evoked by music, which is in line with evidence from previous studies that higher ratings of the valence or emotional intensity of a song are linked to higher familiarity [27, 28, 35, 36] and autobiographical salience [33, 36]. In other words, when music is experienced as familiar and autobiographically salient, it very probably feels also pleasant, arousing, and emotionally intense. The extremely high correlation of emotional intensity to both valence (r = .91) and autobiographical salience (r = .92) suggests

that songs evoking strong emotional experiences are generally experienced as positive [12] and personally meaningful, possibly evoking also feelings of nostalgia [64], which were, however, not assessed here. The pattern of correlations also indicates that the emotional intensity elicited by a song seems to be central for evoking MEAMs, more than the other emotional dimensions, which is also in line with the findings of Janata et al. [33], Schulkind et al. [36], and Talarico et al. [42]. It is possible that the powerful linkage between music-evoked emotions and memories may partly depend on age, as this correlation has been reported to be higher in older adults than in young adults [35, 36].

The correlation between familiarity and autobiographical salience was also very high in the present study (r = .84) compared to the small-moderate correlations reported in previous studies [33, 35, 36]. This may be related to the fact that our subjects were older adults and the song material (traditional folk songs and 1950s to 1970s pop songs) was primarily from the early time period of their life (childhood to young adulthood), which may lead to stronger autobiographical memories. Interestingly, autobiographical salience showed stronger correlations to all emotional ratings than familiarity, suggesting that emotions play a greater role in the evocation of personal memories than the sense of familiarity in music.

The MIR results showed that a set of three musical components—pulse strength, brightness, and low-mid—formed the best explaining models for the valence, emotional intensity, familiarity, and autobiographical salience ratings, accounting for 25%– 46% of the variance in these domains. Arousal, in turn, was explained by pulse strength alone, but only weakly (6%). For the emotional domains, previous MIR studies have reported valence and arousal to be associated with a combination of loudness, tonal, rhythmic, and timbral features [20–22], which is somewhat different than the findings of the present study. For example, features associated with the brightness component have had correlations ranging from small to moderate in some previous studies while being nearly zero (r = -0.04) in our study [21, 22, 65]. These distinctions from previous studies may be related to differences in the sample characteristics, music stimuli, and MIR component model used in the studies. The relative importance of the negative relationship of the pulse strength on all emotional ratings was especially evident in this study and is similar to the observation of Luck et al. [66] that less clear pulse is associated with greater experienced pleasantness. The present results also suggest that the same structural features of music are related both to its emotion-evoking and memory-evoking effect, which makes sense given the very high correlation between the behavioural ratings of the four domains. One interesting matter to point out is that the central musical components of this study—pulse strength, brightness, and low-mid—consisted of mostly short-term features (pulse clarity being the only exception). Some studies have noticed that very short clip of music is enough to form a judgement of familiarity or emotional content of the music [67, 68]. Although this study used averages across the frames, the future studies will hopefully shed more light on the topic. Also concerning musical features (especially low-level features), developing interpretation to a more musically meaningful constructs would be beneficial in the future. Principal component analysis is one possible way to try to address this, but the component structure is not always clear and differs between studies.

Notably, all three musical components had a negative relationship (β value ranging from -0.038 to -0.265) with the behavioural ratings, meaning that songs with a weaker (less clear) pulse, less low middle frequencies and loudness, and less high harmonics and high notes, were generally experienced as more emotion- and memory-evoking. Negative correlations between musical valence and brightness and pulse strength (or similar) features have been reported also in some previous studies [21, 66, 69]. In our study, one plausible explanation for the negative correlations between musical features and behavioural ratings lies in the type of song material and the age of participants. Compared to more contemporary music, old-time music

(traditional folk songs, old waltzes and schlager songs, popular music from 1950s and 1960s), which our participants typically grew up listening to and which was highly represented in our song stimuli, tends to have a weaker pulse and features intermittent tempo slowing. It is also generally characterized by a cleaner, more simple sound, with less bass and high tones and more emphasis on mid tones, and less audio compression and musical post-production. Having become perceptually and emotionally attuned to this type of music in early life, it is possible that elderly listeners find the same features appealing also when hearing newer music. Overall, the observed pattern of musical feature and emotional/memory experience relationships may be specific to older persons and may not be generalizable to younger or middle-age population. In future, it would be interesting to explore if the relationship between musical features and emotional/memory experience is constant or changes with age. Finally, we looked at the potential mediating effects of the emotion variables on the relationship between the memory variables and the musical components. Emotional intensity (and valence to a lesser extent) was found to mediate the predictive effect of the three musical components (pulse strength, brightness, and low-mid) on both autobiographical salience and familiarity, reducing their β values in the model. The mediating effects were larger (full) for low-mid and smaller (partial) for pulse strength and brightness, indicating that even though emotional intensity was the strongest single predictor of autobiographical salience (β = 0.91), pulse strength and brightness still remained significant predictors in the model and, thus, contributed to autobiographical salience above and beyond the effect of emotional intensity.

In conclusion, together with previous studies [33–36], the present results provide compelling evidence that the emotions (especially their intensity) induced by music are intimately linked to its ability to elicit autobiographical memories (MEAMs). The novel finding of the present study is that essentially the same musical features are associated with the emotion- and memory-evoking effects of music in older adults. Generally speaking, there are two possible ways how emotions can influence the strength and amount of MEAMs. First, the emotional experience and its intensity during the moment when a memory is formed may make the memory more vivid and strengthen its consolidation to long-term memory [41–43]. Second, the emotional experience of music when it is heard may make the retrieval of a memory more effective, especially when the music has been associated with a particular emotional experience, which is triggered by hearing the song [36]. In both cases, emotions serve as a strengthening object between music and memories. Of course, it should be kept in mind that the relationship has possible two-way causality: hearing a song that carries strong personal meaning and MEAMs may trigger an emotional response either directly or indirectly by first bringing to mind the memory, which then triggers the emotion [22]. Therefore, the causal aspect of the results should be taken as a suggestion and the full causal structure should remain as a matter of debate. Also, it should be noted that due to the nature of the experiment the stimuli were popular songs containing lyrics. Although this was inevitable, it is possible that the lyrics could have influenced the relationship between musical features and behavioural ratings to some point.

Recent neuroimaging studies have shed some light on the neural mechanisms that link together music, emotions, and autobiographical memory. The key brain areas involved in this seem to be the dorsomedial prefrontal cortex [35, 37] and anterior cingulate [38], which are typically relatively spared in Alzheimer's disease [38], potentially explaining how familiar music can trigger emotions and MEAMs in persons with severe dementia. From a clinical standpoint, the findings of this study could be used to further develop music-based rehabilitation and care practices of elderly people with neurological (e.g., post-stroke aphasia, dementia) or neuropsychiatric (e.g. schizophrenia, severe depression) conditions who cannot communicate their musical preferences, but who could benefit from listening to music as a tool enhance

mood, communication, and cognitive functioning. In these patient populations, knowledge about musical features and their relationship to memory functions and emotional processes could help in optimizing the selection of music for its emotional power, familiarity, and autobiographical salience. For this purpose and for broadening our understanding of music cognition in different clinical groups, it would be interesting and important in future to carry out the present experiment in different neurodegenerative diseases to determine if disease-related changes in emotional processing and cognitive functions influence also the processing of musical features and the emotional and memory-related experience of music.

## Supporting information

**S1 Table. Full list of the 140 songs used in the study.**
(PDF)

**S2 Table. Bivariate correlations between all musical features.**
(PDF)

**S3 Table. Examples songs with highest PCA scores with corresponding musical component and a short description of possible reasons for the high score.**
(PDF)

**S4 Table. Bivariate correlations, regression coefficients, and p-values of familiarity mediation models for each emotion variable.**
(PDF)

**S1 Data. Data used in PCA and regression models.**
(SAV)

## Acknowledgments

We would like to warmly thank the subjects for their participation in the study and the adult education centers, senior citizens' associations, and the choirs of Helsinki, Espoo, and Vantaa region for their generous collaboration in the implementation of the study.

## Author Contributions

**Conceptualization:** Anni Pitkäniemi, Emmi Pentikäinen, Teppo Särkämö.

**Data curation:** Ilja Salakka, Anni Pitkäniemi, Emmi Pentikäinen.

**Formal analysis:** Ilja Salakka.

**Funding acquisition:** Teppo Särkämö.

**Methodology:** Ilja Salakka, Pasi Saari, Petri Toiviainen, Teppo Särkämö.

**Project administration:** Teppo Särkämö.

**Software:** Ilja Salakka, Kari Mikkonen.

**Supervision:** Anni Pitkäniemi, Teppo Särkämö.

**Validation:** Emmi Pentikäinen, Pasi Saari, Petri Toiviainen.

**Visualization:** Ilja Salakka.

**Writing – original draft:** Ilja Salakka, Anni Pitkäniemi, Teppo Särkämö.

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
