## [Decision Letter · Decision Letter 0]

17 Mar 2021

PONE-D-21-03183

What makes music memorable? Relationships between acoustic musical features and music-evoked emotions and memories in older adults

PLOS ONE

Dear Dr. Salakka,

Thank you for submitting your manuscript to PLOS ONE. After careful consideration, we feel that it has merit but does not fully meet PLOS ONE’s publication criteria as it currently stands. Therefore, we invite you to submit a revised version of the manuscript that addresses the points raised during the review process

We look forward to receiving your revised manuscript.

Kind regards,

Stefan Koelsch

Academic Editor

PLOS ONE

Journal Requirements:

'The authors have declared that no competing interests exist.' 

We note that one or more of the authors are employed by a commercial company: Sentina Ltd.

Additional Editor Comments (if provided):

The reviews obtained for this manuscript are from two outstanding experts in the field. Both were positive about your manuscript, and I would be happy if you would consider submitting a revised version of your manuscript. Although the reviewers and I ask only for minor revisions, I would like to ask you to consider each of their very thoughtful comments carefully and address them all in your manuscript (if only as additional points of Discussion, limitation, or further explanation). Also, it would be great if you could make more clear (in the Discussion) in which regard your paper replicates, extends, or contradicts previous findings. I look very much forward receiving a revised version of your manuscript!

Reviewers' comments:

Reviewer's Responses to Questions

**Comments to the Author**

1. Is the manuscript technically sound, and do the data support the conclusions?

Reviewer #1: Yes

Reviewer #2: Partly

2. Has the statistical analysis been performed appropriately and rigorously? 

Reviewer #1: Yes

Reviewer #2: Yes

3. Have the authors made all data underlying the findings in their manuscript fully available?

Reviewer #1: Yes

Reviewer #2: Yes

4. Is the manuscript presented in an intelligible fashion and written in standard English?

Reviewer #1: Yes

Reviewer #2: Yes

5. Review Comments to the Author

Reviewer #1: This is a beautiful study addressing important questions: what makes it that music is memorized in the elderly, and which musical features contribute to memory formation and emotions.

The study is well done and I have only minor points which I have listed:

Introduction:

1.) Second paragraph: I would object that music evoked emotions are universal, definitely they are not, and the two studies quoted (Egermann and Fritz) are painfully euro-centric in defining the "universal emotion" already in the method...please simply put this statement weaker...

2.) Second paragraph: mhh, I see that valence and arousal may be induced by certain musical features, however, these acoustic parameters are in my opinion partly necessary, but not sufficient conditions, they are highly contextual, and every person knows that a sudden change in structure may be extremely arousing, even if this is the contrary of the "parameters" believed to increase arousal. In all of our studies on chills, we only found as a common feature the change in structure, whatsoever this structure was (eg. Grewe et al. 2007, 2009, 2010). Maybe a more "open" formulation would do

3.) Mere exposure: here I missed the beautiful work by Mencke et al. 2019, demonstrating a mere exposure effect even in atonal music

4.) MEAMS: most strongly in the bump of age 14 .. this wonderful paper from the Lyon lab (Kelly Jakubowski et al. 2020) should be given credit.

Methods

1.) Any control, whether the subjects really listened to the excerpts

2.) Any control, whether sound quality was good enough...I see, this is always problematic, Reinhard Kopiez has developed a "check" system for some of his Internet-listening experiments, but this data set here is already collected

3.) The stimuli are mostly songs with text. This is a real problem of all music-emotion studies, that the text is highly important, feelings of nostalgia (as the authors concede), lonesomeness, grief, separation, all these text-components are highly emotion provoking (example Leo Cohen..: https://www.youtube.com/watch?v=NGorjBVag0I)

on the other hand, highly activating classics, like "Rock around the clock" rely much stronger on musical features. This is why we always avoided to include lyrics in the stimuli. I think, this should somehow mentioned, because on the other hand, in favor of the authors argument that musical features are the most important are the results.

Results

1.) I find the results in Fig 1 convincing, concerning the autobiographic salience etc.

2.) I have more problems with the musical features, and I like the idea of doing PCR, however, when coming to the details and contemplating the graphs, there remain some questions open, e.g. that brightness does not contribute to arousal... and that low-mid contributes little (r between -0.26 and 0.01) and that the strongest "memory" pop-ups are pulse-strengths. Therfore, I personally would have preferred to have this more in detailed discussed:

Discussion

1.) As said, I would like to see a more in deep discussion of the parameters emotion provoking characteristics, comparing it with the earlier studies from the Finish and French and German labs...

2.) And maybe some comments on the role of the "song-texts"

3.) The conclusion and the perspectives concerning therapies I like very much!

Reviewer #2: This is a very interesting paper, showing correlational associations among acoustic musical features, emotions and memories in older adults Perhaps the most striking result involves the time frame of the acoustic features—these associations may be revealed in a fraction of a second. However, this result is plausible. The authors might shore up this finding by referencing studies that show that judgments of musical memory and emotion (Krumhansl 2010, Music Perception) and genre (Gjerdingen and Perrott 2010, Journal of New Music Research) are above chance with sound clips of only 300 msec or so. Although these studies deal with young, not older, adults there are related implications for the processes underlying auditory memory and evoked emotion (and visual memory as well).

The authors display a sophisticated understanding of MIR implementation. I have concerns that the behavioral ratings may have been inter-contaminated (a “halo “ effect that produces such high intercorrelations).

The manuscript is well written for the most part; I have some minor editorial recommendations below for improving clarity.

1. Introduction, first paragraph. You might also reference Krumhansl (the 1990 book or the Krumhansl and Cuddy review, 2010) and Rohrmeier (many recent papers) for musical implications of implicit learning of statistical structrures.

2. Introduction, third paragraph. Sentence beginning “Likewise…” has two parts but the two parts are not logically connected.

3. Introduction, final paragraph: “In summary, emotions are one key reason…” Since the data are correlational, I would use the word “reason” with care.

4. Subjects. Why did you choose to assess older adults? Most of the work in this area has been done with university-age subjects. Is that a concern?

5. “subjects [add: had no history of ] neurological or psychiatric disorders”

6. Subjects were 113 adults, but later we learn that only data from the subjects who completed the whole OTMR were included in the analyses. How many subjects, then, were included in the data analysis?

7. Musical Feature Extraction, Paragraph 2 indicates that there was a set of 18 short-term features. However, the number of short-term features in Table 1 is 9. Also, it would help the reader if the order of description of the features in the text and the order in Table 1 was exactly the same. Finally, Table 1 could benefit from some unpacking. For example, explain the “chromagram”. How is strength computed from it? The constructs of key, mode, and tonality have a long history; musically informed readers will want to know precisely how they are handled in the present MIR approach. In addition, the definition of novelty seems vague and therefore difficult for others to follow in a future replication.

8. Consider expanding Table 1 by providing greater explanation of the features in the supplementary materials and, for instructional purpose, provide equations or diagrams.

9. Predicting music-evoked emotions…..(page 11.) The suggesting of linking of components to the sound of percussion instruments is particularly fascinating. Future work might develop the relation between the acoustic features to more musically interpretable constructs.

10. Figure 2 was missing from my file.

11. Mediating effects (page 13 and 14). The logic is clear. Please describe the formatting of Table 4 and the Supplementary Table in more detail. To what regression do the beta values of Table 4 refer?

12. Discussion, (page17, ref. 20). The possible two-way causality is very important and needs to be underscored. The causal direction of the correlational links you have uncovered must remain a matter of debate.

6. PLOS authors have the option to publish the peer review history of their article (what does this mean?). If published, this will include your full peer review and any attached files.

Reviewer #1: **Yes: **Eckart Altenmüller

Reviewer #2: No

---

## [Author Response · Author response to Decision Letter 0]

30 Apr 2021

Thank you for taking our paper to review-process and providing excellent comments from the Editor and the Reviewers. We address first the Editor’s concern about funding/competing interests and provide updated Funding Statement and Competing Interests Statement. Below that we address the Reviewers comments point-by-point.

Reply to editor in respect of updating the Funding Statement and Competing Interests Statement

Thank you for paying attention to the financial disclosure and possible competing interests concerning our study. In this reply we want to underline that although one of the authors had commercial affiliation (Sentina Ltd), it did not profit in any way from the results of this study and there are no competing interests. Also, just to be clear, Sentina Ltd did not provide funding for the study aside paying salary for one of the authors (K.M.) while he was creating the web application which allowed the implementation of the old-time music rating task. Author K.M. who created the web application also participated in commenting the manuscript and helped writing the part where the application and the implementation of the task is described, without any further payments towards Sentina Ltd. The updated Funding Statement and Competing Interests Statement are provided below.

Updated Funding Statement

Financial support for the work was provided by the Academy of Finland (grants 299044, 305264, 306625, and 327996) and the European Research Council (ERC-StG grant 803466). The funders had no role in study design, data collection and analysis, decision to publish, or preparation of the manuscript. Sentina Ltd provided support in the form of salaries for author K.M., but did not have any additional role in the study design, data collection and analysis, decision to publish, or preparation of the manuscript. The specific roles of these authors are articulated in the ‘author contributions’ section.

Updated Competing Interests Statement

The author K.M. is employed by Sentina Ltd. This does not alter our adherence to PLOS ONE policies on sharing data and materials. Sentina Ltd do not profit financially or otherwise from the study results. The authors have declared that no other competing interests exist.

Reply to Reviewer comments

We would like to thank both Reviewers for positive feedback and valuable comments that helped us to improve our manuscript.

Reviewer #1

This is a beautiful study addressing important questions: what makes it that music is memorized in the elderly, and which musical features contribute to memory formation and emotions.

The study is well done and I have only minor points which I have listed:

Response: Thank you very much! We will address all your recommendations point-by-point below.

Introduction:

1.) Second paragraph: I would object that music evoked emotions are universal, definitely they are not, and the two studies quoted (Egermann and Fritz) are painfully euro-centric in defining the "universal emotion" already in the method...please simply put this statement weaker...

Response: We agree with this and have toned down this statement (see page 3, lines 64-65).

2.) Second paragraph: mhh, I see that valence and arousal may be induced by certain musical features, however, these acoustic parameters are in my opinion partly necessary, but not sufficient conditions, they are highly contextual, and every person knows that a sudden change in structure may be extremely arousing, even if this is the contrary of the "parameters" believed to increase arousal. In all of our studies on chills, we only found as a common feature the change in structure, whatsoever this structure was (eg. Grewe et al. 2007, 2009, 2010). Maybe a more "open" formulation would do

Response: We agree that change in any single musical feature (especially in low-level features) is rarely enough to account changes in experienced valence or arousal. However, as we mention in the second paragraph, some features have been previously associated with ratings of valence and arousal (for example based on studies by Eerola et al. 2011, Gingras et al. 2014 and Schubert 2004), meaning that some connection between them has been found.. There may also be a difference between subjective ratings of valence/arousal after exposure to a song and objectively measured time-varying physiological responses (e.g., chills) during exposure to a song, as sudden changes in structure may affect the latter more than the former. We have revised the text slightly to clarify this point (see page 4, lines 72-74).

3.) Mere exposure: here I missed the beautiful work by Mencke et al. 2019, demonstrating a mere exposure effect even in atonal music

Response: Thank you for this tip. We have now added this excellent citation to our manuscript with the notion of atonal music (see page 4, line 91). The following citations in the manuscript have also been renumbered.

4.) MEAMS: most strongly in the bump of age 14 .. this wonderful paper from the Lyon lab (Kelly Jakubowski et al. 2020) should be given credit.

Response: This recommendation is also warmly welcomed, thank you! We have added a citation to this study and it will probably also support the writing of our next paper from this data where our focus will be on the time era of the songs (see page 5, lines 110-111).

Methods

1.) Any control, whether the subjects really listened to the excerpts

Response: Since the subjects performed the music listening and rating task in the home setting, at their own time and pace, we have no way of objectively ascertaining that they indeed listened to each excerpt. However, given that (i) the task started with a short practice session, in which the subjects were able to test the user interface and set the volume to a comfortable but clearly audible level and (ii) each song excerpt played automatically through once before the subject were able to answer the first question, we are fairly confident that the subjects performed the task as instructed. This clarification has been added to the manuscript (see page 8, lines 156-157). Overall, the subjects participated in the study voluntarily without any monetary compensation and based on the informal feedback we received they were generally motivated and enjoyed doing the task.

2.) Any control, whether sound quality was good enough...I see, this is always problematic, Reinhard Kopiez has developed a "check" system for some of his Internet-listening experiments, but this data set here is already collected

Response: We agree that controlling for sound quality is always difficult in an online study performed at home. We had no objective check system in place, but we sought to minimize the risk of poor audio quality by (i) asking the subjects to use headphones or an external speaker when performing the task, (ii) providing and tablet computer and speaker to those subjects who did not have their own equipment, (iii) including an practice trial with volume setting before the trial, and (iv) instructing the subjects to contact one of the authors in case they had any sound issues or other technical problems.

3.) The stimuli are mostly songs with text. This is a real problem of all music-emotion studies, that the text is highly important, feelings of nostalgia (as the authors concede), lonesomeness, grief, separation, all these text-components are highly emotion provoking (example Leo Cohen..: https://www.youtube.com/watch?v=NGorjBVag0I)

on the other hand, highly activating classics, like "Rock around the clock" rely much stronger on musical features. This is why we always avoided to include lyrics in the stimuli. I think, this should somehow mentioned, because on the other hand, in favor of the authors argument that musical features are the most important are the results.

Response: Thank you for this very good point. It is true that most of our song stimuli included lyrics, and we realize that lyrics are also important in eliciting emotions in music. We considered this issue when planning the study, but opted to include songs with lyrics in order to increase the ecological validity of the stimuli and their chances of being familiar and eliciting autobiographical memories to subjects (limiting the stimuli e.g. to instrumental classical or jazz music would have reduced this aspect in many subjects). However, while lyrics may have influenced the emotional (and memory) experience of music to some extent, the pattern of associations between ratings and musical features still emerged in the results with these stimuli, as noted by the Reviewer. The possible effect of lyrics on the results is now mentioned in the discussion (see page 21, lines 420-422).

Results

1.) I find the results in Fig 1 convincing, concerning the autobiographic salience etc.

Response: Thank you!

2.) I have more problems with the musical features, and I like the idea of doing PCR, however, when coming to the details and contemplating the graphs, there remain some questions open, e.g. that brightness does not contribute to arousal... and that low-mid contributes little (r between -0.26 and 0.01) and that the strongest "memory" pop-ups are pulse-strengths. Therfore, I personally would have preferred to have this more in detailed discussed:

Response: We originally made the decision not to discuss all the smaller relationships to 1) keep the focus on the main results and 2) since this is exploratory study, small correlations might not contain much evidence for relationships. However, the point that absent or almost absent relationships (especially the ones that were present in previous studies) should be discussed in some way is good and is addressed below.

Discussion

1.) As said, I would like to see a more in deep discussion of the parameters emotion provoking characteristics, comparing it with the earlier studies from the Finish and French and German labs...

Response: We have now added some more discussion accounting the absence of relationship between brightness and arousal and raising the relative importance of the pulse strength component on the emotional ratings (see page 19, lines 366-372). From our viewpoint it is important (in the context of the present study) to integrate the discussion about emotional effects of acoustic features to musical memories (MEAMs and familiarity). In the paragraph starting “Notably, all three musical components had a negative relationship…” (pages 19-20, lines 383-398), we discuss these relationships in the context of memory effects to a greater extent.

2.) And maybe some comments on the role of the "song-texts"

Response: As mentioned above, the possible effect of lyrics is now discussed briefly as a limitation of the study.

3.) The conclusion and the perspectives concerning therapies I like very much!

Response: Thank you very much for this and for all other thoughtful comments to improve our manuscript!

Reviewer #2

This is a very interesting paper, showing correlational associations among acoustic musical features, emotions and memories in older adults Perhaps the most striking result involves the time frame of the acoustic features—these associations may be revealed in a fraction of a second. However, this result is plausible. The authors might shore up this finding by referencing studies that show that judgments of musical memory and emotion (Krumhansl 2010, Music Perception) and genre (Gjerdingen and Perrott 2010, Journal of New Music Research) are above chance with sound clips of only 300 msec or so. Although these studies deal with young, not older, adults there are related implications for the processes underlying auditory memory and evoked emotion (and visual memory as well).

The authors display a sophisticated understanding of MIR implementation. I have concerns that the behavioral ratings may have been inter-contaminated (a “halo “ effect that produces such high intercorrelations).

The manuscript is well written for the most part; I have some minor editorial recommendations below for improving clarity.

Response: Thank you for the overall positive feedback and very good recommendations!

Thank you for raising the point about frame lengths into discussion. This is an interesting point overall. Unfortunately, we think that it is not plausible to use the results of our study as an evidence for the short time frame associations between stimuli and behavioural responses. This is because although the time frames were used to extract features were short (0.025 sec and 3 sec for short and long term features, respectively) they were averaged over the whole song excerpt before any of the further analyses. This removes the possibility to infer simple associations between ratings and time frames. However, this is an important topic and we have added these references you pointed out to the manuscript (page 19, lines 374-379). Thank you for the references!

By the “halo” effect we think that the Reviewer means that if the listener likes the song he/she rates it higher in every behavioural rating independent of does he/she actually think or feel that way (e.g. listener rates a song high in arousal just because he/she finds it pleasant even though it did not actually raise the listeners arousal state). We think this is possible to some point but unlikely with most of the correlations of the present study. In the Discussion (page 18, lines 341-360) we explain our view of these high intercorrelations and in our view most of them are in line with previous studies. For example, emotional intensity had highest correlations to valence (r = .91) and autobiographical salience (r = .92). These are indeed strong correlations for behavioural research, but also quite plausible since positive musical emotions are usually stronger than negative ones (Zentner et al. 2008) and relationship between autobiographical saliency and emotional intensity is both intuitive and has been observed before (e.g. Schulkind et al. 1999). 

Other recommendations are addressed point-by-point below.

1. Introduction, first paragraph. You might also reference Krumhansl (the 1990 book or the Krumhansl and Cuddy review, 2010) and Rohrmeier (many recent papers) for musical implications of implicit learning of statistical structrures.

Response: Thank you for these excellent recommendations. We have now added these citations to the part concerning implicit learning by exposure to music (see page 3, line 51).

2. Introduction, third paragraph. Sentence beginning “Likewise…” has two parts but the two parts are not logically connected.

Response: We see this as a logical structure, since both parts of the sentence are essentially telling the same thing: Familiarity of music correlates with pleasantness of music. To explain this: The first part of the sentence tells that the familiarity of music and liking of it are connected. The second part deepens that statement telling that repetition (getting to know the music better) increases chances of the music being rated more emotionally intense and more pleasurable.

3. Introduction, final paragraph: “In summary, emotions are one key reason…” Since the data are correlational, I would use the word “reason” with care.

Response: The Reviewer is right in that the word “reason” is probably too strong in this context. We have weakened the wording of this statement (see page 5, line 112).

4. Subjects. Why did you choose to assess older adults? Most of the work in this area has been done with university-age subjects. Is that a concern?

Response: This study is originally part of larger longitudinal study where we investigate the neurocognitive effects of senior choir singing during ageing. Given the increasing interest on music as a tool to support healthy ageing and in dementia care, we thought it is important and novel to focus on this age group, especially since MIR studies have not been done in this population before. One can also argue that the link between emotions and autobiographical memories evoked by music may be particularly evident in older adults given their longer musical and life event experience, and therefore it makes sense to study the phenomenon in this population. While this approach detracts from the generalizability of the findings (not applicable across life), it adds to their specificity. Moreover, our future plan is to continue the study also in clinical populations (age-related neurodegenerative diseases), for which the present data will provide a direct comparison. In the Discussion, we already note that the “observed pattern of musical feature and emotional/memory experience relationships may be specific to older persons and may not be generalizable to younger or middle-age population”. We have now added a sentence suggesting that the effects of age could be addressed in a future study (see page 20, lines 397-398).

5. “subjects [add: had no history of ] neurological or psychiatric disorders”

Response: Thank you for this comment. This clarification is added to the manuscript (see page 7, line 138).

6. Subjects were 113 adults, but later we learn that only data from the subjects who completed the whole OTMR were included in the analyses. How many subjects, then, were included in the data analysis?

Response: All the 113 subjects completed the whole OTMR which resulted in no missing values or no subjects removed prior to data analysis. There seemed to be a partially misleading part of sentence concerning this in the Methods. We have now removed this confusing sentence (from page 9, lines 186-187). Thank you for noticing this.

7. Musical Feature Extraction, Paragraph 2 indicates that there was a set of 18 short-term features. However, the number of short-term features in Table 1 is 9. Also, it would help the reader if the order of description of the features in the text and the order in Table 1 was exactly the same. Finally, Table 1 could benefit from some unpacking. For example, explain the “chromagram”. How is strength computed from it? The constructs of key, mode, and tonality have a long history; musically informed readers will want to know precisely how they are handled in the present MIR approach. In addition, the definition of novelty seems vague and therefore difficult for others to follow in a future replication.

Response: Concerning the amount of short-term features in Table 1, Sub-band fluxes include 10 different sub-bands, each of which are treated as individual features. The numbers 1-10 are now added to the Table 1 (after the “Sub-band fluxes”) to underline this.

It is a good point that the same order of features in Table 1 and in the text would make it easier to read. This is now done as far as possible within the limits of time windows in Table 1 (RMS energy and Novelty do not fit in any of the other categories and RMS energy is a short-term feature and Novelty is a long-term-feature).

We have also added to the notes below the Table 1 the more detailed description of chromagram and key strength. Please see also the response to the next (8.) comment. Also, more precise formulation of the Novelty was added to the Table 1.

8. Consider expanding Table 1 by providing greater explanation of the features in the supplementary materials and, for instructional purpose, provide equations or diagrams.

Response: This is very good recommendation. To address this we have added following line to the paragraph before Table 1 “More thorough technical descriptions of the features can be found from Lartillot (2017).”, which refers to the MIRtoolbox manual (see page 9, line 195). The MIRtoolbox manual contains all features described in technical but easily understandable manner. We ended up to this decision because every feature used in this study is already well described in that manual (which is freely available at https://www.jyu.fi/hytk/fi/laitokset/mutku/en/research/materials/mirtoolbox/manual1-7-2.pdf).

9. Predicting music-evoked emotions…..(page 11.) The suggesting of linking of components to the sound of percussion instruments is particularly fascinating. Future work might develop the relation between the acoustic features to more musically interpretable constructs.

Response: Thank you! The low-mid component in this study is likely to be linked with some percussion instruments (and of course also pulse strength component at least to bass drum). This is very interesting topic. We have added a sentence to the Discussion about the possible future prospects about this topic (page 19, lines 379-381).

10. Figure 2 was missing from my file.

Response: We are sorry for this. We are quite confident that Figure 2 was included in the pdf we checked after the submission, but maybe there have been some technical issues. We promise to double check this in the revised submission.

11. Mediating effects (page 13 and 14). The logic is clear. Please describe the formatting of Table 4 and the Supplementary Table in more detail. To what regression do the beta values of Table 4 refer?

Response: We have now made the legend of Table 4 (and S4 Table) more clear, explaining a bit more thoroughly the different components of the tables.

12. Discussion, (page17, ref. 20). The possible two-way causality is very important and needs to be underscored. The causal direction of the correlational links you have uncovered must remain a matter of debate.

Response: It is true that the two-way causality needs to be taken into account well enough. To underline this more, we have added a sentence (page 21, lines 419-420) to remind that the causality of the results should be taken as a suggestion and the whole causal structure should remain as a matter of debate. Thank you for this important note and for all other great recommendations!

---

## [Editor Report · Decision Letter 1]

3 May 2021

What makes music memorable? Relationships between acoustic musical features and music-evoked emotions and memories in older adults

PONE-D-21-03183R1

Dear Dr. Salakka,

We’re pleased to inform you that your manuscript has been judged scientifically suitable for publication and will be formally accepted for publication once it meets all outstanding technical requirements.

Kind regards,

Stefan Koelsch

Academic Editor

PLOS ONE
---

## [Editor Report · Acceptance letter]

6 May 2021

PONE-D-21-03183R1 

What makes music memorable? Relationships between acoustic musical features and music-evoked emotions and memories in older adults 

Dear Dr. Salakka:

I'm pleased to inform you that your manuscript has been deemed suitable for publication in PLOS ONE. Congratulations! Your manuscript is now with our production department. 

Kind regards, 

on behalf of

Prof. Dr. Stefan Koelsch 

Academic Editor

PLOS ONE